# Neuroimmune Crosstalk in Rheumatoid Arthritis

**DOI:** 10.3390/ijms23158158

**Published:** 2022-07-24

**Authors:** Dashuang Gao, Xu Gao, Fan Yang, Qingwen Wang

**Affiliations:** 1The Brain Cognition and Brain Disease Institute, Shenzhen Institute of Advanced Technology, Chinese Academy of Sciences, Shenzhen-Hong Kong Institute of Brain Science-Shenzhen Fundamental Research Institutions, Shenzhen 518055, China; ds.gao@siat.ac.cn; 2University of Chinese Academy of Sciences, Beijing 100049, China; 3Shenzhen Key Laboratory of Inflammatory and Immunology Diseases, Shenzhen 518036, China; 2211110779@stu.pku.edu.cn; 4Department of Rheumatism and Immunology, Peking University Shenzhen Hospital, Shenzhen Peking University-Hong Kong University of Science and Technology Medical Center, Shenzhen 518036, China

**Keywords:** rheumatoid arthritis, affective disturbance, sensory nerve, sympathetic nervous system, parasympathetic nervous system

## Abstract

Recent studies have demonstrated that immunological disease progression is closely related to abnormal function of the central nervous system (CNS). Rheumatoid arthritis (RA) is a chronic, inflammatory synovitis-based systemic immune disease of unknown etiology. In addition to joint pathological damage, RA has been linked to neuropsychiatric comorbidities, including depression, schizophrenia, and anxiety, increasing the risk of neurodegenerative diseases in life. Immune cells and their secreted immune factors will stimulate the peripheral and central neuronal systems that regulate innate and adaptive immunity. The understanding of autoimmune diseases has largely advanced insights into the molecular mechanisms of neuroimmune interaction. Here, we review our current understanding of CNS comorbidities and potential physiological mechanisms in patients with RA, with a focus on the complex and diverse regulation of mood and distinct patterns of peripheral immune activation in patients with rheumatoid arthritis. And in our review, we also discussed the role that has been played by peripheral neurons and CNS in terms of neuron mechanisms in RA immune challenges, and the related neuron-immune crosstalk.

## 1. Introduction

RA patients have a significantly higher incidence of physical disability and psychiatric comorbidity than the general population, by which we can analyze the relationships between psychological factors and RA, and the relationships between the mechanisms of psychological and physiological factors that take part in these comorbidities [1,2,3]. Neurons and the immune system exhibit bidirectional interactions and play key roles in organ homeostasis, immune responses, and regulation of inflammation. The nervous system can receive stimulation from immune cells, and at the same time, the signals of the CNS act on immune cells through the peripheral nervous system (PNS) to regulate the inflammatory response [4,5]. In addition, PNS interacts with immune cells in the gut, lungs, skin, and other peripheral organs to form neuroimmune cell units that initiate inflammatory states and maintain tissue homeostasis. In addition, inflammatory mediators such as nerve growth factor, C-C Motif Chemokine Ligand, and pro-inflammatory cytokines, can also interact with receptors in joint nociceptors then inducing their activation and sensitization in RA [6]. Neuron-derived neuropeptides and neurotransmitters that neurons can produce modulate the function and response of immune cells, while immune cells can also produce inflammatory factors that increase neuronal activity. A growing body of research and clinical evidence suggests that peripheral nerve endings and immune cells in peripheral organ tissues are co-localized in local tissues and interact to regulate each other [7,8,9]. The various cytokines, inflammatory factors, neuromodulators, and neurotransmitters produced by nerve cells and immune cells regulate each other, promoting this interaction and feedback loop between immune cells and central, peripheral, sympathetic, and parasympathetic nerves [4,10]. Therefore, through this mini-review, we reviewed the interaction between nerve and immunity during the occurrence and development of RA, trying to analyze the relationship between RA and factors such as psychology, pathology, inflammation, etc., and explore the interaction of neuroimmune and the molecules that regulate inflammation through RA. Based on this review, neuroimmune crosstalk plays a key role in the pathophysiology of immune diseases and neuroimmune interactions regulate the pathophysiology of RA, through direct and indirect mechanisms.

## 2. Affective Disturbance in RA

### 2.1. Psychiatric Comorbidities

Compared with the general population, RA patients are known to endure a higher chance of developing psychiatric comorbidities including depression [11] and anxiety, mood, and psychotic disturbances [12], which greatly reduce the quality of life and physical and mental health. Compared with people without physical defects or recurring pain [13], mental health disorders in RA patients are associated with harmful effects such as pain, fatigue, impaired sleep quality, increased mental health-related distress, and passive pain-coping strategies [14,15]. Psychiatric comorbidities also adversely affect multiple outcomes in RA patients [16], including poor medication adherence and treatment response (Figure 1) [17]. For example, RA patients with depression are associated with an inferior response to biological therapy [18].

Depression and anxiety disorders are relatively well-recognized and studied psychiatric disorders in RA patients. However, due to many reasons, the estimated prevalence of these two differs rather largely from study to study, the incidence of anxiety and depression in RA patients appears to range from 26% to 46% and 14.8% to 34.2%, respectively [19]. In addition, the etiology behind it has not been adequately studied [20].

### 2.2. Depression

Among the related psychiatric disorder in RA, depression is the most closely associated one [21]. However, estimates of the prevalence of depression are influenced by many factors, including differences in measurement methods, recurrence of depressive symptoms, and variances in disease duration [19]. In addition, there are some overlapping diagnostic symptoms existing both in RA and depression, such as fatigue, poor appetite, and sleeping disturbance, thus some studies have removed items that may overlap between the two diseases or adjusted diagnostic thresholds, inevitably affecting the prevalence statistics of RA patients and depression [22].

Nonetheless, the evidence for the negative impacts of depression on the course of RA is convincing. Notably, previous studies have shown that depression accounts for 6.9% of mortality in RA patients [23] and is associated with increased mobility in RA, impaired quality of life, reduced chance of remission and increased arthritis-related complications [24]. The feelings of hopelessness and non-adherence to treatment caused by depression can also further aggravate the disease, which may explain why patients with moderate, severe, or people with chronic depression are less likely to follow medical guidance than people with mild or no depression, regardless of the medical condition [25]. For example, people with rheumatoid arthritis who suffer from chronic depression are less likely to take their doctor’s advice and are less likely to take disease-modifying antirheumatic drugs (DMARDs) as prescribed than patients with milder or subclinical levels of depression. According to the inflammatory hypothesis, depression may also be responsible for the increase in inflammation and disease activity in RA [26].

In recent years, several studies have also found a bidirectional relationship between depression and rheumatoid arthritis. Notably, in addition to the high incidence of depression in RA patients, people with depression are at greater risk of developing rheumatoid arthritis than the general population [27], and antidepressants are reported to have a protective and therefore confounding effect on RA [28]. In addition, excessive activation of the hypothalamic-pituitary-adrenal (HPA) axis due to cytokine activity has also been associated with depression [29].

### 2.3. Anxiety

The prevalence of psychological comorbidities in RA patients has been extensively researched, particularly depression and anxiety [20]. Several studies have noted that anxiety is more common than depression in people with rheumatoid arthritis [30]. The prevalence of anxiety in RA patients, however, differs across research from 20% to 26% of patients classified as having obvious anxiety disorders or anxiety-like symptoms [31,32]. In addition, depressed RA patients are more likely to experience symptoms of anxiety compared with general RA patients or age-matched individuals [20].

Anxiety in RA patients is related to multiple factors, such as age, sex, socioeconomic status, pain, marital status, and disease activity [32,33,34]. Anxiety symptoms in RA patients are linked to a lower quality of life [35] and poor therapeutic response, which are similar to depression. For example, persistent anxiety in RA patients can reduce treatment response to prednisolone by 50% [36].

Research on anxiety shares common problems with depression, such as small populations with various prevalence estimates [19] and overlapping diagnostic symptoms. Regarding the impact of RA on patient anxiety, studies have reported that c-reactive protein (CRP) levels may be correlated with anxiety and depression levels in RA patients [34]. However, it is still unclear if RA increases the prevalence of anxiety independently or if this association is confounded by the anxiety tendencies that are already existed [37], and thus more research is needed.

### 2.4. Schizophrenia

In comparison to the general population, patients with RA have a rather low prevalence of schizophrenia, according to clinical and epidemiological studies [38]. The risk of RA in schizophrenic patients is estimated to be 30–50% of the control [39]. One common theory to explain this is the abnormal inflammatory cytokine profile in schizophrenia patients, which includes circulating proinflammatory cytokines [40], elevated levels of soluble IL-2 receptor, aberrant tryptophan metabolism, prostaglandin insufficiency, and therapy side effects that are linked to RA-schizophrenia [41]. As noted previously, high blood concentrations of IL-1 receptor antagonists may have protective effects against RA in schizophrenic patients [42].

Genetics is another important factor in both diseases. Notably, based on the well-studied human leucocyte antigen (HLA) system [43], HLADRB1*0101, HLA-DRB1*04 (*0401, *0403, *0405, and *0406), and HLA class II antigens and alleles have been implicated in the regulation of antibody-mediated immune responses [44]. HLA-DRB1, including HLA-DR4 serotypes, positively correlated with RA incidence and negatively correlated with schizophrenia levels [45]. Therefore, it could be assumed that the negative genetic correlation between RA and schizophrenia may be due to the different roles of variants in immune response pathways in different tissues and/or in response to different challenges [41].

### 2.5. Bipolar Disorder

Although autoimmune diseases are associated with psychiatric disorders, especially depression and anxiety, little research has been done on the relationship between rheumatoid arthritis and bipolar disorder [46]. In addition, asthma, cirrhosis, and alcoholism are important risk factors for developing bipolar disorder [47].

Research has also suggested that peripheral chronic inflammation is a key factor in the pathophysiology of CNS inflammation in bipolar disorder [48]. Thus, RA is a disease characterized by long-term chronic inflammation, and the process of inflammatory immune response activated by RA may affect the prognosis and development of bipolar disorder.

Sex-related characteristics have also been found to play a key role in developing bipolar disorder in RA patients. Bipolar disorder is equally prevalent in men and women, according to epidemiological studies [49]. Female RA patients are more likely to acquire bipolar disorder compared to women without RA. This may be due to sex steroid hormones, which play an important regulatory role in immune responses; specifically, estrogen-induced immune stimulation in RA may be more likely to trigger inflammation [50]. However, further clinical and basic research is needed on the relationship between RA and bipolar disorder.

## 3. CNS Modulation of RA

The CNS is assumed to be protected by the blood-brain barrier (BBB) from inflammatory signals in the circulatory system. The injury of CNS would result in the disturbance of normal and balanced interaction between immune system and CNS by interfering a variety of biological processes, such as the sympathetic nervous system (SNS), PNS and HPA axis [51]. Furthermore, CNS involvement in RA is considered unusual compared to other autoimmune diseases such as lupus erythematosus [52]. In recent years, among the neurological sequelae associated with chronic inflammatory diseases, including RA, it has been reported that chronic peripheral inflammation is closely related to the central nervous system [53], with CNS involvement, e.g., meningitis, rheumatoid nodules, and cerebral vasculitis, as also have been reported [54]. In addition, RA patients may have CNS comorbidities such as dementia, demyelination, malignancy, stroke, or inflammation, all of which may be affected by long-term systemic immune activation and drug therapy [54]. For example, glucocorticoids and non-steroidal anti-inflammatory medicines (NSAIDs) may promote the occurrence of cerebral ischemia, but hydroxychloroquine treatment may prevent stroke [55]. It has long been speculated that adaptive immune responses in lymphoid organs may be directly controlled by brain activity.

A recent study identified a neural circuit in which CRH neurons regulate spleen immune cells from the paraventricular nucleus (PVN) [56]. In addition, the number of antibody-secreting cells produced by mice following splenic nerve removal after vaccination decreased sharply, indicating that impulse signals of the splenic nerve may promote humoral responses [56]. These findings imply that a central-peripheral nerve circuit directly regulates lymphocyte-mediated adaptive immune responses, which could be the biological basis for immune response behavioral modulation.

The IL-1 signaling pathway is important for a variety of pathologies in CNS and is an important proinflammatory cytokine involved in innate immunity. IL-1 levels in the plasma concentrations in RA patients are much greater than that in the general population, which is also linked to the severity of the disease [57]. IL-1 levels in synovial fluid are also linked to several radiological and histological characteristics of RA [58]. Furthermore, the neuronal type I IL-1 receptor (nIL-1R1) can utilize different coreceptors for signal transduction, IL-1 excites neurons without activating the transcription of inflammatory cytokines in the nuclear factor kappa B (NF-κB) pathway [59]. Studies have also shown that IL-1A + polymorphisms may affect RA risk in the overall population, while IL-1B+ polymorphisms may affect RA risk in the general and Asian populations [60]. However, how IL-1 participates in RA pathogenesis via the CNS and potential treatments require further study.

IL-1A+ polymorphisms have also been linked to an increased incidence of RA in the general population, while IL-1B+ polymorphisms may affect RA risk in the general and Asian populations [60]. However, how IL-1 participates in RA pathogenesis via the CNS and potential treatments require further study [54]. The anti-TNF drugs and rituximab have been reported in the treatment of bacterial and polyoma virus-induced brain infections [61]. For non-anti-TNF drugs, existing RA cohort data are not persuasive due to their small sample size and short exposure time. Non-anti-TNF medicines, like anti-TNF therapies, may help protect the CNS from direct and indirect harm due to systemic inflammation. Biological agents, on the other hand, have been linked to a few special cases of CNS neoplasms, such as lymphoma, and can also promote the development of opportunistic CNS infections. Research also indicates that anti-TNF agents may worsen the symptoms of demyelinating disorders [54].

In conclusion, neurological symptoms in patients with RA should be closely monitored, and treatment programs should be altered accordingly. In addition, as symptoms may vary widely, long-term treatment follow-up is also recommended [54].

## 4. The Regulatory Role of the Peripheral Nervous System in the Development of RA

The PNS regulates immune cell function and redistributes energy to the immune system. During an acute immune response, peripheral nerve endings undergo rapid reorganization and changes in response to the immune response [62]. Four different PNS types affect the inflammatory response: sensory nervous system, which is the most complex nervous system in the PNS and acts mainly in proinflammation; peripheral sympathetic nervous system (SNS), the main neurotransmitter is NA (noradrenaline) and NPY (nerve peptide Y), SNS has both pro-inflammatory and anti-inflammatory effects; parasympathetic nerves mainly play a role in suppressing inflammation. The main neurotransmitter in the parasympathetic nervous system is acetylcholine, the anti-inflammatory and pro-inflammatory effects of PNS are mainly achieved through the α7 subunit of the nicotinic acetylcholine receptor (α7nAChR) which can be damaged by RA. The PNS plays multiple roles in inflammation. Different components of PNS will show different pro-inflammatory and anti-inflammatory effects according to the tissue of the inflammatory response, different nerve fiber densities, and inflammatory response time [62].

PNS plays an active and broad role in RA inflammation. For example, peripheral nerve endings direct the regulation of immune effector cells and modulate the immune system’s energy supply. In the early stage of acute inflammation, systemic sympathetic nerve activity increases, while systemic parasympathetic nerve activity decreases. In the later stage, sympathetic nerve fibers at the site of inflammation decrease, the innervation of sensory nerve fibers increases, and sensory nerve fiber activity increases [63]. Therefore, the sympathetic, parasympathetic, and sensory nerves of the PNS all have pro- and anti-inflammatory effects, depending on the site of inflammation, the density of nerve fibers, and the stage of inflammation.

### 4.1. Sensory Nervous System Interactions with Immune Cells in RA

Sensory neurons not only provide the brain with a bridge to changes in the external world but also sense changes in internal organs and tissues. The important function of the sensory nervous system is to sense, transmit and process sensory information. Sensory nervous system activation is an inevitable part of the immune response, with sensory immune information sent to regions of the brain and spinal cord to mount an appropriate response. Receptors on sensory nerve endings are the first step in the recognition of external “intruders”, which can directly activate or lower the activation threshold of the receptor [64,65].

In the immune response, sensory neurons differ in their sensitivity to immune stimulation, conduction velocity, and neuropeptide production. Peripheral sensory nerves dominate the body’s barrier system and internal organs. Including the skin, mucous membranes, and organs, sensory nerve endings signal directly to immune cells, directly regulating their function. At the same time, the peripheral sensory nervous system expresses multiple sensors, including transient receptor potential (TRP) channels, purine energy P2X channels, mechanically sensitive ion channels, G protein-coupled receptors (GPCR), and cytokine receptors, that directly detect inflammatory and mediators and foreign invaders [4]. Once activated, these sensory nerves send pain signals to the central nervous system and release neuropeptide and non-peptide media from their terminals, effectively modulating acute and chronic inflammatory responses in immune cell activation (Figure 2). Transient receptor potential A1 (TRPA1) is a transient receptor potential ion channel that causes calcium ions to enter cells when activated. Sensory neurons expressing TRPV1 are responsive to high temperature and capsaicin, and TRPA1 ion channels are sensitive to chemical and mechanical stress, chemical stimuli such as mustard, and low temperatures [66,67]. In bacteria-infected mice, neurons expressing Nav1.8 can release CGRP to inhibit the recruitment of immune cells. Ablation of Nav1.8 and TRPV1 will result in the inability of dendritic immune cells to produce IL-23 and reduce the production of IL-17 by immune cells, thereby acting as an anti-inflammatory that reduces the recruitment of inflammatory cells and reduces subsequent inflammation [68]. TRPM8 a receptor activated by cold has been implicated in menthol and cold-induced hypersensitivity reactions [69]. Due to the high diversity of peripheral sensory neurons, further work is needed to determine how different subsets of neurons interact with immune cells.

Sensory nerve activation is part of the immune response. As the first step to recognizing immune response, sensory nerves can directly activate or lower the threshold of receptor activation to achieve signal perception [70]. This activation eventually leads to the local release of neurotransmitters and neuropeptides such as the substance P (SP) and calcitonin gene-related peptide 1 (CGRP-1), which exhibit potent vasodilatory and chemotactic properties leading to innate and adaptive immune cell recruitment [71]. Injection of SP or CGRP-1 into the skin upregulated the endothelial cell adhesion molecule e-selectin, followed by rapid recruitment of eosinophils and neutrophils within 15 min. These neurotransmitters and sensory neuropeptides act directly on different immune cells and establish a pro-inflammatory environment (i.e., neurogenic inflammation) by promoting the extravasation of plasma proteins. These sensory mechanisms enhance local immune responses positively, it reflects the important role of sensory nerves as part of the basic “alarm system” of the human body.

Currently, the role of MS in collagen-induced arthritis (ASD) has yet to be studied. However, SP is known to have a pronounced pro-inflammatory effect on adjuvant arthritis (AIA), highlighting the general proinflammatory effect of the sensory nervous system on arthritis [72,73]. In rats, intraarticular administration of an SP receptor antagonist three days after AIA induction reduces hyperalgesia and cartilage loss [73,74].

Many cytokine families provide specific functions. Cytokines are produced mainly by immune cells, but other central nervous system cells also produce cytokines. The most studied cytokines in psycho-neuroimmunology are IL-6, tumor necrosis factor (TNF-a), IL-1β, and interferon (IFN). Pro-inflammatory cytokines induce receptor activator of NF-κB ligand (RANKL), prostaglandins, and matrix metalloproteinases that mediate RA symptoms such as pain, swelling, cartilage, and bone degradation. Stimulated by IFN, TNF, and IL-6, osteoclasts are produced in the synovial membrane, promoting bone damage. Sialic acid, as a very important molecule in neurodevelopment, has also been shown to be related to RA in recent years. Sialic acid may be a potential biomarker of RA, which deserves further attention [75]. These molecular and cellular events translate into clinical manifestations of the disease, and the progression of joint lesions is intrinsically linked to joint swelling. Cytokines are small proteins that affect cell function and interaction and have pro-inflammatory or anti-inflammatory properties. Under physiological conditions, peripheral immune cells do not enter the brain parenchyma, although some cells are present in the cerebrospinal fluid and meningitis. However, under certain conditions, macrophages and T cells can cross the blood-brain barrier (BBB) and enter the brain parenchyma, often causing damage. The BBB is made up of endothelial cells, which bind together at tight junctions, limiting access to immune cells, various blood components, and pathogens. The BBB prevents over 98% of antibodies and small molecules from entering the parenchymal cells while ensuring other molecules are effluxed. Several hypotheses have been proposed to explain how peripheral immune cells cross BBB under pathological conditions. In environments where the BBB is weakened, or where BBB is more permissive, immune cells penetrate the brain’s parenchyma through the barrier. Studies have shown that cytokines can directly affect the central nervous system by acting on the nervous system through the BBB barrier, but current research has focused on the immunomodulation of peripheral nerves.

IL-6 plays an important role in systemic inflammatory and arthritic disease [76]. In particular, mice reduce in IL-6 exhibit a significant reduction in the AIA model [77]. Moreover, in mouse models of TNF-mediated human inflammation, IL-6 promotes osteoclast formation from inflammation and bone remodeling [78]. IL-6 binds to IL-6 membrane receptors (IL-6R) or IL-6 soluble receptors (sIL-6R), enabling IL-6 delivery in cells that do not express membrane receptors. Finally, the IL-6-IL-6R complex is connected to the transmembrane subunit of gp130 signaling [79]. In chronic models of arthritis, administration of sgp130 may reduce inflammation.

IL-1β is widely expressed in RA disease [80,81]. Mice that do not have a natural IL-1 inhibitor will have a spontaneous development of erosive arthritis. Disabling IL-1β may reduce the immune response in the CIA model, but not in the IAA model. Bone loss was increased when RA was induced by the CIA and AIA models [82]. Treatment with anakinra, an IL-1 receptor antagonist (IL-1R1), shows limited efficacy in humans [80]. IL -1β binds to the IL-1R1 and IL-1R1 receptors on the cell surface. The IL-1R1 sensors transmit the biological signal from IL-1β to the cells and the IL-1RII serves as a decoy receiver [83,84]. Interestingly, in rats, about 25% of all small and medium DRG neurons express IL-1R1, this proportion of neurons increases to 60% when immunized with AIA models and persists in AIA [82]. IL-1β may also be involved in RA-induced pain and hypersensitivity, for example, plantar injection of IL-1β in rats induces skin pain, hypersensitivity, and transient discharges [85,86].

Recent studies have shown that IL-17 plays an important role in regulating immunity and inflammation in rheumatoid arthritis and other autoimmune diseases [80]. IL-17A is the prototypic element and induces innate and adaptive immune mediators [87,88]. The study found that in RA patients, the level of IL-17A in synovial effusion was significantly increased [88], and decreased or increased expression of IL-17A can ameliorate or aggravate disease activity, respectively [88]. Previous studies have shown that monoclonal antibodies to IL-17 can effectively reduce the immune response to human autoimmune diseases [87,89].

Given the effects of TNF-α on neurons, neutralization of TNF-α is known to reduce pain in animal models and patients with RA. Notably, neutralization of TNF in various murine models of arthritis (e.g., CIA [90], AIA [91], and arthritis [92]) can rapidly reduce pain and allergic reactions caused by inflammation, in the absence of other analgesics. Typically, TNF-neutralizing anti-noxious effects were observed on the first day of treatment [91,93]. Similarly, RA patients treated with infliximab may experience a significant reduction in pain post-day 1 [94]. Intraarticular injection of etanercept may decrease the response of joint nociceptors to inflamed joint rotation within 30 min, suggesting that neutralizing TNF-α in the joint is a significant contributor to therapeutic effects [91].

### 4.2. The Sympathetic Nervous System in RA

One of the main functions of the sympathetic nervous system is to coordinate stress responses and prepare the body for fight or flight in stressful or dangerous situations to protect itself from possible harm. Most tissues, including lymph nodes and spleen, are dominated by SNS fibers. The immune system and nervous system are not two separate systems, and there is a close dialogue and communication between them. This dialogue and exchange play a vital role in the health and disease of living organisms. As the initial warning system, the sensory nervous system activates the response system acts as the primary of the HPA axis, and the SNS transports the information from the CNS to the periphery. The cell bodies of the preganglionic neurons are distributed along the spinal cord, and their axons terminate on the cell bodies of sympathetic neurons in the paravertebral or prevertebral ganglia. Most postganglionic neurons use norepinephrine as their primary neurotransmitter; therefore, sympathetic neuron-immune cell regulation is carried out primarily through the norepinephrine pathway. Physiologically, dynamic changes in sympathetic activity and norepinephrine levels have been confirmed in lymphoid tissue in response to systemic stressors including endotoxemia and bacterial infections. Second, immune cells mostly express adrenergic receptors, including macrophages, dendritic cells, T lymphocytes, B lymphocytes, natural killer cells, and congenital lymphocytes (ILCs), making them sensitive to norepinephrine pathways. Synergistic activation of these two systems allows the body to fight against foreign antigens [62] and redistribute energy resources to the immune system for activation and use [95,96]. However, the CNS needs to distinguish between local inflammation, which requires little energy to resolve, and whole-body systemic inflammation (e.g., sepsis), which requires tremendous energy to resolve. Neuroendocrine energy redistribution in response to immune activation may be initiated by direct neural activation through the sensory system and by circulating cytokines such as IL-6 [97]. Cytokines in the circulatory system are a natural “sensor” of inflammation levels and energy distribution in the body, while activation of the SNS and HPA axis, which control cortisol levels, leads to the release of fatty acids, amino acids, and glucose, which activate immune cells in organs such as the spleen and lymph nodes [98,99,100].

In recent decades, with the use of animal models of arthritis, such as the AIA in Lewis rats and the AIA in DBA1 mice, the characterization of the role of PNS in arthritis disease has improved. Research has concentrated on the role of the SNS, using chemical (selective catecholaminergic neurotoxin 6-hydroxydopamine), pharmacological (β-adrenoceptor antagonists), and immunological (antibodies against nerve terminals) approaches. Studies in the AIA rat model focused on innate immune function and T-cell responses rather than humoral immunity, suggesting that SNS mediates pro-inflammatory effects through a beta 2-adrenergic mechanism [101,102]. However, when the SNS is damaged in the draining lymphatic outcome, the SNS again shows anti-inflammatory properties. The dual role of the sympathetic nerve depends on where it interacts with immune cells. Similar complexity has been observed at the CIA, which relies more on humoral immunity and B cells than the AIA model. At the CIA, depending on the point in time of sympathetic nephrectomy, the SNS plays a dual role, acting as an anti-inflammatory at an early stage and an anti-inflammatory at a later stage [103].

Norepinephrine (NA), the main neurotransmitter of the SNS, inhibits the proliferation of lymphocytes and the release of proinflammatory cytokines, e.g., IL-2, IL-12 [104], interferon, and TNF-α, through beta-2 adrenergic receptors [62]. NA also stimulates anti-inflammatory cytokine production (IL-4, IL-5, and IL-10) [105]. Not only postganglionic fibers but also lymphocytes NA and acetylcholine are released, thereby regulating immune function in an autocrine and paracrine manner. The concentration of neurotransmitters released by symptoms is determined by the number of vesicles released (depending on the speed of release) and the density of nerve dominance. Both components are dynamic and can change during inflammatory processes. Initial inflammation results in increased sympathetic nerve activity and increased local neurotransmitter release [63]. Sympathetic neurotransmitters such as NA (high concentration) and co-transmissible adenosine showed potent inhibition of immune cells through β2-adrenaline receptor and adenosine α2 receptor-mediated signaling [106,107].

The SNS controls various physiological processes, such as vascular tension, cardiac output, glandular secretion, and fever. When the environment changes, the sympathetic nervous system maintains homeostasis by controlling these physiological processes, regulating the activity of end organs as well as cell tissues mainly by releasing various neurotransmitters from local axons. In real life, when the work pressure is high, the psychological burden is heavy, and the emotions are tense, people are often prone to illness. What is the reason? This is how the sympathetic nervous system affects the immune system. Due to the complexity of immune cell innervation and the emergence of emerging populations of the sympathetic nerve, molecular mechanisms of bidirectional communication between sympathetic nerves and target cells have not been elucidated.

### 4.3. Parasympathetic Nervous System

The parasympathetic nervous system is often thought to be the opposite of the SNS. This feature is correct in the early stages of inflammation when parasympathetic activity is primarily directed at fighting energy supply and attenuating inflammation throughout the body. However, it is not clear if the parasympathetic nervous system directly or indirectly affects immune cells or the activation of SNS. The parasympathetic nervous system innervates the body’s internal organs primarily through fibers originating in the dorsal motor nucleus and the brainstem fuzzy nucleus and is projected to the internal organs of the chest and abdomen through the vagus nerve, whereas the preganglionic and postganglial fibers of the vagus nerve can be activated by cytokines, inflammatory products, and molecular ligands from pathogens.

Current studies have shown that the anti-inflammatory effect of vagus nerve stimulation depends on the activation of postgangliocacinergic neurons carried in the spleen nerve, and the spleen nerve activates T cells expressing choline acetyltransferase (ChAT) through β2-AR so that these T cells synthesize and release ACh, which in turn acts on macrophages through α7nAChR to inhibit the inflammatory cytokine response. Interestingly, inhibition of the parasympathetic nervous system is not only dependent on α7nAChR expression [108] but also the expression of β2-adrenoceptors in CD4+CD25 lymphocytes [108,109]. To ensure sufficient inflammation intensity, parasympathetic activity is suppressed during the initial inflammatory process [110,111]. For example, experimental stimulation of the peripheral vagus nerve following a proximal incision strongly suppresses acute inflammation caused by lipopolysaccharides [111], which depends on sympathetic stimulation of the vagus nerve of the spleen [112]. Leukocyte recruitment is also inhibited by vagus nerve stimulation. As a result, these results suggest that the parasympathetic nervous system is anti-inflammatory in the first phase of inflammation [113].

However, little is known about the role of the parasympathetic nervous system in the later stages of chronic inflammation. The parasympathetic nervous system and its main neurotransmitter, acetylcholine, appear to pass through the α7nAChR during lipopolysaccharide-induced acute inflammation. However, the role of the parasympathetic nervous system in models of antigen-dependent RA is unknown. CIA development can be inhibited using α7nAChR-specific agonists [114]. The findings above contradict each other, showing that the severity of arthritis increases and decreases separately in the same model. Therefore, it is difficult to summarize the specific effects of the parasympathetic nervous system on chronic arthritis in current studies. Interpretation of the data is complex and diverse, as it has not been observed in RA that peripheral immune activity is mediated by the output parasympathetic nervous system, rather than directly affecting immune cells by humoral hormones, or indirectly by activating the SNS. Therefore, the parasympathetic nervous system is not further elaborated on in this review.

Meaningful discussion of the interplay between RA physiological and psychological processes requires an understanding of the many dimensions in which this interaction occurs. The nervous system plays a monitoring and regulating role in the inflammatory process. The sensory nervous system acts as an “early warning system,” monitoring and notifying other parts of the body of ongoing local inflammation. The sensory nervous system activates SNS in the spinal segments and the hypothalamus. PNS plays a key role in regulating the body’s defense mechanisms and coordinating the immune response. Therefore, it is crucial to understand the different drivers that control the temporal and spatial regulation of nervous system responses. Understanding the interactions between immunity, endocrinology and the nervous system opens the avenue for potential drugs aiming to modulate inflammatory pathological processes and improve prognosis. Therefore, a patient’s adaptation to RA must be understood in the context of the patient’s overall society, as the presence of interpersonal stress and support can have short-and long-term effects on health, coping strategies, and treatment responses. Further understanding of the interaction between the nervous system and the peripheral immune system will have positive implications for future RA treatments and drug development.

## Figures and Tables

**Figure 1 ijms-23-08158-f001:**
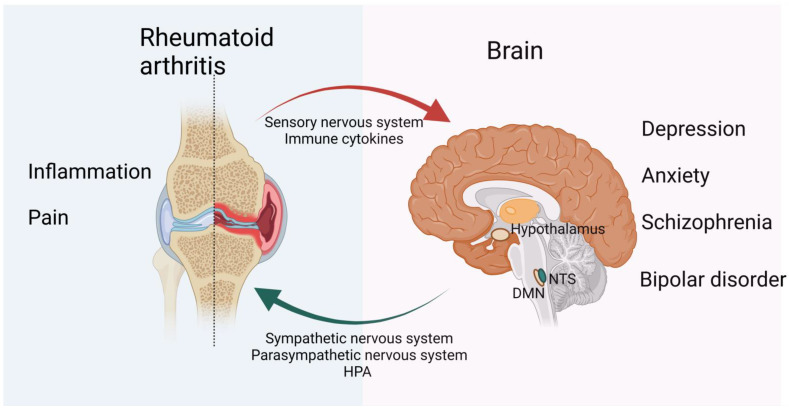
Bidirectional psychological and neurological effects of RA and the brain on each other. Recurrent pain and prolonged inflammation are prominent symptoms of RA and are strongly associated with mental illnesses such as depression and anxiety. In contrast, mental health can influence RA disease activity and is associated with reduced treatment response in RA patients. For example, patients with depression have an increased risk of RA, while antidepressants are reported to have a protective and therefore confounding effect on RA. The CNS and PNS both play a role in inflammation. The vagus nerve is the main efferent pathway that mediates immunosuppression of the CNS. It controls the production of TNF and other proinflammatory cytokines through the splenic nerve. Sensory nerves are activated by proinflammatory cytokines, such as IL-1 and IL-6, and sensory immune information is then sent to regions of the brain and spinal cord to mount an appropriate response. The HPA axis and SNS then carry information from the CNS to the PNS. While the parasympathetic nervous system is anti-inflammatory in the first phase of inflammation, its role in the later phases of chronic inflammation requires further research. Figure created using BioRender (https://biorender.com, accessed on 20 July 2022).

**Figure 2 ijms-23-08158-f002:**
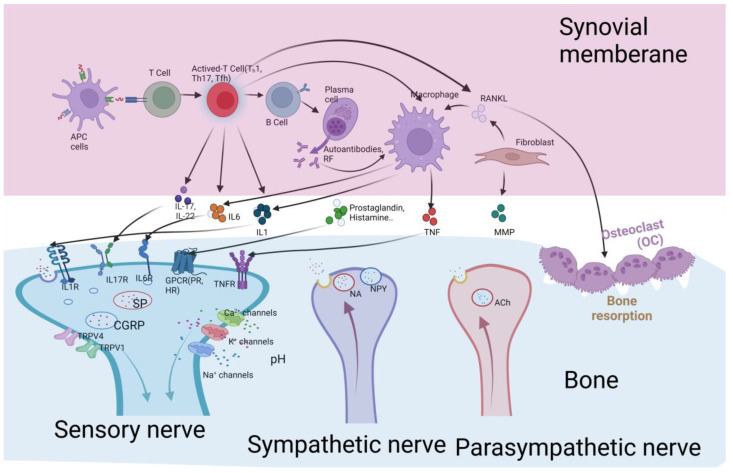
Molecular mechanisms underlying immune-neuron interface. Schematic: Sensory, sympathetic, and parasympathetic nerve fibers innervate immune cells during inflammation in RA. Rheumatoid arthritis is characterized by synovial infiltration of multiple joints, including T cells, B cells, and monocytes. The expansion of synovial fibroblast-like cells and macrophage-like cells led to the proliferation of the synovial lining layer. This dilated synovium, invades the bony cartilage joints around the joints, causing bone erosion and cartilage degeneration. During this process, activated T cells, B cells, plasma cells, macrophages, and fibroblasts release a variety of immune factors, such as IL-1, IL-6, IL-17, IL-22, and TNF. With joint cartilage injury, sensory nerve endings infiltrate the joint. Sensory nerves express IL-1R, IL-6R, IL-17R, TNFR, and other receptors, which sense joint lesions to transmit information to the dorsal root ganglia or the CNS, resulting in the release of CGRP, SP, and other neuropeptides. Sympathetic and parasympathetic nerves are also present in the joints. The main neurotransmitters of the sympathetic nerve are NA and NPY, and the sympathetic nerve is a single and late inflammatory effect in the early stage of inflammation. The main transmitter of the parasympathetic nerve, acetylcholine, acts to inhibit inflammation. IL-1 (interleukin-1), IL-6 (interleukin-6), IL-17 (interleukin-17), NA (noradrenaline), NPY (nerve peptide Y), Ach (acetylcholine), CGRP (calcitonin gene-related peptide), SP (substance P), TRPV1 (transient receptor potential vanilloid1), and TRPV4 (transient receptor potential vanilloid 4). Figure created using BioRender (https://biorender.com, accessed on 20 July 2022).

## Data Availability

Not applicable.

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
