# Peer review of "Neuroimmune Crosstalk in Rheumatoid Arthritis"

_ijms, 2022, doi:10.3390/ijms23158158_

Round 1
Reviewer 1 Report
This paper study the relationship between an inflammatory disease ( rheumatoid arthritis ) and neurologic/psychiatric disease. Both diseases are relevant and prevalent. This topic is interesting. The analysis of pathophysiological mechanism is very interesting and this allows new therapeutic options. The topic is original. It´s unknow to many clinicians. The topic stablishes the basis of the relationship between both diseases. The paper is well written, clear and eay to read.The conclusions are clear and consistent with the presented evidence. The paper can accept to the current form. The clasification minor is due to change the indicated word.
Minor concerns
- Page 4, line 136 change morbility to mobility
Author Response
Reviewer 1:
Page 4, line 136 change morbility to mobility
Responses: Thank you for the comments, we have change morbility to mobility.(Page 4, line 142)
Reviewer 2 Report
The manuscript (ID: ijms-1825675) entitled “Neuroimmune crosstalk in rheumatoid arthritis” has been submitted by Dashuang Gao, Xu Gao, Fan Yang and Qingwen Wang as a review article to the IJMS section: Molecular Immunology - Inflammation in the CNS and PNS: From Molecular Basis to Therapy. This draft documents a solid piece of scientific work and will be of high interest for the general readership of this section of the journal.
The authors should discuss the contents of two further articles in their review article.
Vasconcelos DP, Jabangwe C, Lamghari M, Alves CJ. The Neuroimmune Interplay in Joint Pain: The Role of Macrophages. Front Immunol. 2022 Mar 10;13:812962. doi: 10.3389/fimmu.2022.812962. PMID: 35355986; PMCID: PMC8959978.
Ying-ying Huang, Xueli Li, Xiaojin Li, Yuan-yuan Sheng, Peng-wei Zhuang, Yan-jun Zhang (2019) Neuroimmune crosstalk in central nervous system injury-induced infection and pharmacological intervention. Brain Research Bulletin. Volume 153, Pages 232-238.
https://doi.org/10.1016/j.brainresbull.2019.09.003.
Much more important: the authors have ignored the impact of sialic acid completely which play a crucial role in rheumatoid arthritis but also in many cases of mental disorders.
Li W, Liu Y, Zheng X, Gao J, Wang L, Li Y. Investigation of the Potential Use of Sialic Acid as a Biomarker for Rheumatoid Arthritis. Ann Clin Lab Sci. 2019 Mar;49(2):224-231. PMID: 31028068.
The authors should close this gap.
